# Never Go Full Batch
# (in Stochastic Convex Optimization)

**Idan Amir**
Department of EE
Tel Aviv University
idanamir@mail.tau.ac.il

**Yair Carmon**
Blavatnik School of CS
Tel Aviv University
ycarmon@tauex.tau.ac.il

**Tomer Koren**
Blavatnik School of CS, Tel Aviv U
and Google Research
tkoren@tauex.tau.ac.il

**Roi Livni**
Department of EE
Tel Aviv University
rlivni@tauex.tau.ac.il

## Abstract

We study the generalization performance of *full-batch* optimization algorithms for stochastic convex optimization: these are first-order methods that only access the exact gradient of the empirical risk (rather than gradients with respect to individual data points), that include a wide range of algorithms such as gradient descent, mirror descent, and their regularized and/or accelerated variants. We provide a new separation result showing that, while algorithms such as stochastic gradient descent can generalize and optimize the population risk to within $\varepsilon$ after $O(1/\varepsilon^2)$ iterations, full-batch methods either need at least $\Omega(1/\varepsilon^4)$ iterations or exhibit a dimension-dependent sample complexity.

## 1 Introduction

Stochastic Convex Optimization (SCO) is a fundamental problem that received considerable attention from the machine learning community in recent years [28, 15, 4, 11, 2]. In this problem, we assume a learner that is provided with a finite sample of convex functions drawn i.i.d. from an unknown distribution. The learner's goal is to minimize the expected function. Owing to its simplicity, it serves as an almost ideal theoretical model for studying generalization properties of optimization algorithms ubiquitous in practice, particularly *first-order methods* which utilize only first derivatives of the loss rather than higher-order ones.

One prominent approach for SCO—and learning more broadly—is to consider the *empirical risk* (the average objective over the sample) and apply a first-order optimization algorithm to minimize it. The problem of learning is then decoupled into controlling the optimization error over the empirical risk (training error) and bounding the difference between the empirical error and the expected error (generalization error).

In convex optimization, the convergence of different first-order methods has been researched extensively for many years (e.g., [26, 25, 5]), and we currently have a very good understanding of this setting in terms of upper as well lower bounds on worst-case complexity. However, in SCO where the generalization error must also be taken into account, our understanding is still lacking. In fact, this is one of the few theoretical learning models where the optimization method affects not only the optimization error but also the generalization error (distinctively from models such as PAC learning and generalized linear models). In particular, it has been shown [28, 15] that some minima of the empirical risk may obtain large generalization error, while other minima have a vanishingly small

generalization error. To put differently, learning in SCO is not only a question of minimizing the empirical risk, but also a question of *how* one minimizes it. However, the results of [28, 15] leave open the question of whether concrete optimization also have different generalization properties.

Towards better understanding, Amir et al. [2] recently studied the generalization properties of full-batch gradient descent (GD), where each step is taken with respect to the gradient of the empirical risk. For GD (and a regularized variant thereof), they gave a lower bound on the generalization error as a function of iteration number, which is strictly larger than the well-known optimal rate obtained by stochastic gradient descent (SGD), where each step is taken with respect to the gradient at a sampled example. Notably, the lower bound of [2] precisely matches the dimension-independent stability-based upper bound recently shown for full-batch GD by Bassily et al. [4]. The separation between full-batch GD and SGD is the first evidence that not only abstract Empirical Risk Minimizers may fail to generalize in SCO, but in fact also basic methods such as GD could be prone to such overfitting. A natural question is, then, whether overfitting is inherent to *full-batch* algorithms, that minimize the objective only through access to the exact empirical risk, or whether this suboptimality can be remedied by adding regularization, noise, smoothing, or any other mechanism for improving the generalization of GD.

In this work we present and analyze a model of *full-batch* optimization algorithms for SCO. Namely, we focus on algorithms that access the empirical risk only via a first-order oracle that computes the *exact* (full-batch) gradient of the empirical loss, rather than directly accessing gradients with respect to individual samples. Our main result provides a negative answer to the question above by significantly generalizing and extending the result of Amir et al. [2]: we show that *any* optimization method that uses full-batch gradients needs at least $\Omega(1/\varepsilon^4)$ iterations to minimize the expected loss to within $\varepsilon$ error. This is in contrast with the empirical loss, which can be minimized with only $O(1/\varepsilon^2)$ steps.

Comparing SGD and GD in terms of the sample size $n$, we see that SGD converges to an optimal generalization error of $O(1/\sqrt{n})$ after $O(n)$ iterations, whereas a full-batch method must perform $\Omega(n^2)$ iterations to achieve the same $O(1/\sqrt{n})$ test error. We emphasize that we account here for the *oracle complexity*, which coincides with the iteration complexity in the case of gradient methods. In terms of individual gradients calculations, while SGD uses at most $O(n)$ gradient calculations (one sample per iteration), a full-batch method will perform $\Omega(n^3)$ calculations (*n* samples per iteration).

The above result is applicable to a wide family of full-batch learning algorithms: regularized GD (with any *data-independent* regularization function), noisy GD, GD with line-search or adaptive step sizes, GD with momentum, proximal methods, coordinate methods, and many more. Taken together with upper bound of Bassily et al. [4], we obtain a sharp rate of $\Theta(1/\varepsilon^4)$ for the generalization-complexity of full-batch methods. Surprisingly, this rate is achieved by standard GD (with an unusual step-size choice of $\eta = \Theta(\varepsilon^3)$), and it cannot be improved by adding regularization of any sort, nor by adding noise or any other form of implicit/explicit bias.

## 1.1 Related work

This work extends and generalizes the results of Amir et al. [2] who proved generalization lower bounds for GD (and a specific instance of regularized GD). Our work shows that in fact *any* full-batch method will suffer from similar lower bounds. Our construction builds upon the one used in [2], which in turn builds upon previous constructions [4, 28]. However, our arguments and proofs here are more challenging, as we need to reason about a general family of algorithms, and not about a specific algorithm whose trajectory can be analyzed directly. Our developments also build on ideas from the literature on oracle complexity lower bounds in optimization [25, 26, 30, 8, 12, 9]. In particular, we first prove our result in the simplified setting of algorithms constrained to the span of observed gradients [25, 26] and subsequently lift it to general algorithms using a random high-dimensional embedding technique proposed by Woodworth and Srebro [30] and later refined in [8, 12]. However, while these works lower bound what we call the empirical risk, we lower bound the generalization error. This requires us to develop a somewhat different argument for how the span of the gradients evolve during the optimization: in prior work, the algorithm learns the component of the solution coordinate by coordinate, whereas in our work the true (generalizing) solution is present in the observed gradients from the first query, but spurious sampling artifacts drown it out.

Empirical studies (outside of the scope of SCO) support the claim that generalization capabilities degrade with the increase of the batch size. Specifically, Zhu et al. [33] indicates that SGD outperforms GD in terms of generalization. The works of Keskar et al. [22] and Hoffer et al. [20] exhibit a similar phenomenon in which small-batch SGD generalizes better than large-batch SGD with the same iteration budget. We provide the first *theoretical* evidence for this phenomenon for *convex* losses. Several theoretical studies explore the convergence of stochastic methods that use mini-batches [10, 23, 31]. Note that this setting differs from ours, as they assume access to mini-batches sampled *without replacement* whereas full-batch means we reuse the same (full) batch with each gradient step.

There has also been recent progress in improving the generalization capabilities of GD. Wu et al. [32] interprets mini-batch SGD as a noisy version of GD. They propose a modified algorithm with noise injected to the full-batch gradients. Geiping et al. [16] propose a GD-based training scheme that achieves CIFAR-10 generalization performance comparable to standard SGD training. Interestingly, both proposed algorithms require access to sample-points and are therefore not "full-batch" by our definition: The scheme [32] requires sample-point data for computing the noise, while the GD variant [16] uses mini-batch statistics to compute a regularization term (as well as batch normalization). Our work shows that (in SCO) this is unavoidable: namely, no data-independent noise or full-batch regularization can be used to improve generalization at a reasonable computational budget.

Several other works study the generalization performance of GD [29, 17, 21, 24]. The work of Soudry et al. [29], for example, examines GD on unregularized logistic regression problems. They show that, in the limit, GD converges to a well-generalizing solution by arguing about the bias of the algorithm. Interestingly, both our and their results require slow-training, beyond what is required for empirical error optimization. Another work that highlights the slow convergence of GD is that of Bassily et al. [4]. They were the first to address uniform stability of (non-smooth) GD and SGD, and provided tight bounds. Stability entails generalization, hence our results lead to stability lower bounds for any full-batch method. Consequently, we extend the lower bounds for GD in the work of Bassily et al. [4] to a wider class. It might be thought that the instability argument of Bassily et al. [4] can be used to obtain similar generalization lower bounds—however, we note that their techniques also prove instability of SGD (which does generalize). Hence, instability does not immediately imply, in this setting, lack of generalization.

Finally, we note that under smoothness and strong convexity, it is well known that improved rates can be obtained. Specifically, using the stability bound of Bousquet and Elisseeff [6] one can show that we can achieve generalization error of $O(1/\sqrt{n})$ after $O(n)$ iterations if the population risk is $O(1)$-strongly convex. The arguments of Hardt et al. [19] imply generalization bound to instances where every sample risk is $O(\sqrt{n})$ smooth. Our result implies that, even though these special families of functions enjoy appealing learning rates, in general it is impossible to obtain better rates by strong-convexifying or smoothing problem instances via first-order full-batch oracle queries.

## 2 Problem Setup and Main Results

We study the standard setting of stochastic convex optimization. In this setting, a learning problem is specified by a fixed domain $\mathcal{W} \subseteq \mathbb{R}^d$ in $d$-dimensional Euclidean space, and a loss function $f : \mathcal{W} \times \mathcal{Z} \to \mathbb{R}$, which is both convex and $L$-Lipschitz with respect to its first argument (that is, for any $z \in \mathcal{Z}$ the function $f(w; z)$ is $L$-Lipschitz and convex with respect to $w$). In particular, throughout the paper, our construction consists of 1-Lipschitz functions and we will focus on a fixed domain $\mathcal{W}$ defined to be the unit Euclidean ball in $\mathbb{R}^d$, namely $\mathcal{W} = \{w : \|w\|_2 \le 1\}$.

We also assume that there exists an unknown distribution $D$ over parameters $z$ and the goal of the learner is to optimize the *true risk* (or *true loss*, or *population risk*) defined as follows:

$$F(w) := \mathop{\mathbb{E}}_{z \sim D} [f(w; z)], \tag{1}$$

We assume that a sample $S = \{z_1, \dots, z_n\}$ is drawn from the distribution $D$, and the learner has to output $w_S \in \mathcal{W}$ (the exact access the learner has to the sample, and how $w_S$ may depend on $S$ is discussed below). We require the solution to be $\varepsilon$-optimal in expectation for some parameter $\varepsilon > 0$, i.e.,

$$\mathop{\mathbb{E}}_{S \sim D^n} [F(w_S)] - \min_{w^\star \in \mathcal{W}} F(w^\star) \le \varepsilon.$$

As discussed, the standard setting assumes that the learner has direct access to the i.i.d. sample, as well as to the gradients of the loss function (i.e., a first-order oracle). In this work, though, we focus on a specific family *full-batch* methods. Hence, the optimization process is described as follows: First, an i.i.d. sample $S = (z_1, \ldots, z_n)$ is drawn from $D$. Then, the learner is provided with access only to the *empirical risk* via a *full-batch first-order oracle* which we define next.

**Full-batch first-order oracle.** Consider a *fixed* sample $S = (z_1, \ldots, z_n)$ of size $n$, drawn i.i.d. from $D$. The *empirical risk* over the sample $S$ is

$$F_S(w) = \frac{1}{n} \sum_{i=1}^{n} f(w; z_i).$$

Then, a *full-batch first-order oracle* is a procedure that, given input $w \in \mathcal{W}$, outputs

$$\mathcal{O}(w) := (\nabla F_S(w); F_S(w)).$$

where $\nabla F_S(w)$ is an empirical risk sub-gradient of the form

$$\nabla F_S(w) = \frac{1}{n} \sum_{i=1}^{n} \nabla f(w; z_i), \tag{2}$$

and each sub-gradient $\nabla f(w, z_i)$ is computed by the oracle as a function of $w$ and $z_i$ (that is, independently of $z_j$ for $j \neq i$).

We emphasize that the sample is fixed throughout the optimization, so that the oracle computes the gradient of the *same* empirical risk function at every call, hence the name *full-batch*. Note that the subgradient with respect to a single data point, i.e., $\nabla f(w; z_i)$, is not accessible through this oracle, which only returns the average gradient over the sample $S$.

Notice that our definition above is slightly narrower than a general sub-gradient oracle for the empirical risk due to the requirement that the sub-gradients $\nabla f(w, z_i)$ are chosen independently of $z_j$ for $j \neq i$ – since we provide here with a lower bound, this restriction strengthens our result. We make this restriction to avoid some degenerate constructions (that in fact can even be used to fail SGD if the gradient at $z_i$ may depend on the whole sample), which are of no practical implications.

**Full-batch first-order algorithm.** A *full-batch* (first-order) method is naturally defined as any algorithm that has access to the optimization objective—namely the empirical risk $F_S$—only via the full-batch first order oracle. In particular, if $w_t$ is the $t$'th query of the algorithm to the full-batch oracle then $w_t$ has to be of the form

$$w_t = Q_t(\mathcal{O}(w_0), \ldots, \mathcal{O}(w_{t-1})), \tag{3}$$

where $Q_t : (\mathbb{R}^{d+1})^t \to \mathcal{W}$ is a fixed (possibly randomized) mapping. At the end of the process the algorithm outputs $w_S$. We study the algorithm's oracle complexity, which is the number of iterations $T$ the algorithm performs before halting. Therefore, we assume without loss of generality that $w_S = w_T$, i.e., the algorithm's output is its $T$'th query.

## 2.1 Main result

In this section we establish our main result, which provides a generalization lower-bound for full-batch first order algorithms. The complete proof is provided in the full version of the paper [1].

**Theorem 2.1.** *Let $\varepsilon > 0$ and $n, T \in \mathbb{N}$; there exists $d = poly(2^n, T, 1/\varepsilon)$ such that the following holds. For any full-batch first-order algorithm with oracle complexity at most $T$, there exists a 1-Lipschitz convex function $f(w; z)$ in $\mathcal{W}$, the unit-ball in $\mathbb{R}^d$, and a distribution $D$ over $\mathcal{Z}$ such that, for some universal constant $c > 0$:*

$$\mathbb{E}_{S \sim D^n} [F(w_S)] \geq \min_{w^\star \in \mathcal{W}} F(w^\star) + \varepsilon + \Omega\left(\min\left\{1 - c\varepsilon^2\sqrt{T}, 0\right\}\right). \tag{4}$$

An immediate consequence of Theorem 2.1 is that in order to obtain less than $\varepsilon$ true risk we need at least $T = \Omega(1/\varepsilon^4)$ iterations.

For simplicity, we state and prove the lower bound in Theorem 2.1 for the class of first-order full-batch algorithms defined above. However, our constructions readily generalize to *local full-batch oracles* that provide a complete description of $F_S$ in an arbitrarily small neighborhood of the query point [25, 18]. Such oracles subsume second-order oracles, and consequently our generalization lower bounds hold also for second-order full-batch algorithms.

## 2.2 Discussion

Theorem 2.1 suggests that full-batch first-order algorithms are inferior to other types of first-order algorithms that operate with access to individual examples, such as SGD. Importantly, this separation is achieved not in terms of the *optimization* performance but in terms of the *generalization* performance. In light of this result, we next discuss and revisit the role of the optimization algorithm in the context of SCO. In particular, we wish to discuss the implications to what are perhaps the two most prominent full-batch optimization methods, GD and regularzied-GD, and in turn compare them.

**Gradient descent.** Perhaps the simplest example of a full-batch method is (projected) GD: GD is an iterative algorithm that at each iteration performs an update step

$$w_t = \Pi_{\mathcal{W}}[w_{t-1} - \eta \nabla F_S(w_t)],$$

where $\mathcal{W}$ is a convex set on which we project the iterated step. The output of GD is normally taken to be $w_S = \frac{1}{T} \sum w_t$ (or a randomly chosen $w_t$). Notice, that each step requires one call to a full batch oracle, and a single projection operation. The convergence analysis of GD to the optimal solution of the *empirical risk* has been widely studied. Specifically, if $n$ is the sample-size, it is known that with $\eta = O(1/\sqrt{n})$ and $T = O(n)$, GD converges to a minimizer of $F_S$ that is $O(1/\sqrt{n})$-sub optimal. For the *exact* variant of GD depicted above, the generalization performance was analyzed in the work of Amir et al. [2] that showed that with $T = O(n)$ steps, GD will suffer $\Omega(1/\sqrt[4]{n})$ generalization error. Theorem 2.1 extends the above result to any variant of GD (dynamic learning-rate, noisy GD, normalized GD, etc.).

**Regularized gradient descent.** We would also like to discuss the implication of Theorem 2.1 with respect to *regularized* variants of GD that operate on the regularized empirical risk

$$\hat{F}(w) = \lambda r(w) + F_S(w).$$

The main motivation of introducing the regularization term $r$ is to avoid overfitting, and a popular choice for $r$ is the Euclidean norm $r(w) = \|w\|_2^2$. This choice leads to the following update rule for GD:

$$w_{t+1} = \Pi_{\mathcal{W}} \left[ (1 - \eta_t) \cdot (2\lambda w_t) - \eta_t \nabla F_S(w_t) \right],$$

Again, this update can be implemented using a single first-order full-batch oracle call that computes the quantity $\nabla F_S(w_t)$. More generally, for any data-independent $r$, GD on $\hat{F}$ is a full-batch algorithm[1]. When $r$ is the Euclidean norm, the minimizer of $\hat{F}$ is known to enjoy (with choice $\lambda = O(1/\sqrt{n})$), an optimal generalization error of $O(1/\sqrt{n})$ [6, 28]. This demonstrates the power of regularization and how it can provably induce generalization. Nevertheless, Theorem 2.1 still applies to any optimization method over $\hat{F}$. Since optimization of $\hat{F}$ (the regularized empirical risk) to $O(1/\sqrt{n})$-precision can be done via a full-batch method, and with less than $O(n)$ calls, we observe that there are methods that minimize the regularized-empirical risk but, due to Theorem 2.1 do not reach the optimal generalization error.

**The role of regularization.** Finally, in light of Theorem 2.1 let us compare the different variants of GD and regularized GD that *do* generalize well, in order to sharpen our understanding of the role of regularization in generalization. The conclusion of Theorem 2.1 is that any full-batch method that generalizes well performs at least $O(n^2)$ steps. For regularized GD, with $\ell_2$ regularization, $O(n^2)$ are indeed sufficient. In particular, with $O(n^2)$ iterations we can find a solution that has $O(1/n)$ empirical error. Any such solution would enjoy a generalization error of $O(1/\sqrt{n})$ [28]. For GD, Bassily et al. [4] showed that $O(n^2)$ iterations would also suffice to achieve $O(1/\sqrt{n})$ error. This is achieved by tuning the learning rate to $\eta = O(1/n^{3/2})$. Notice that this improvement does not require any type of added regularization.

To summarize, both GD and regularized GD with optimal parameters require $\Theta(n^2)$ iterations to attain the optimal $O(1/\sqrt{n})$ generalization error. Overall then, explicitly adding regularization is not necessary nor does it improve the convergence rate. One might be tempted to believe that tuning the learning rate in GD induces *implicitly* some sort of regularization. For example, one might

---

[1]Note that we are not concerned with the computational cost of computing $\nabla r(w_t)$ since it does not factor into oracle complexity.

imagine that GD can be biased towards minimal norm solution, which might explain redundancy of regularizing by this norm. However, this turns out also to be false: Dauber et al. [11] showed how GD (with any reasonable choice of learning rate) can diverge from the minimal norm solution. In fact, for any regularization term $r$, one can find examples where GD does not converge to the regularized solution. Thus, even though GD and regularized-GD are comparable algorithms in terms of generalization and oracle complexity, they are distinct in terms of the solutions they select.

## 3 Technical Overview

In this section we give an overview of our construction and approach towards proving Theorem 2.1. For the sake of exposition, we will describe here a slightly simpler construction which proves the main result only for algorithms that remain in the span of the gradients. In more detail, let us examine the family of iterative algorithms of the form

$$w_t \in \text{span}\{\nabla F_S(w_0), \nabla F_S(w_1), \ldots, \nabla F_S(w_{t-1})\} \cap \mathcal{W}, \tag{5}$$

where $\mathcal{W}$ is the unit ball and $\nabla F_S(w_t)$ is full-batch oracle response to query $w_t$ as defined in (2) above. Well-studied algorithms such as GD and GD with standard $\ell_2$ norm regularization fall into this category of algorithms.

To extend the lower bound to algorithms not restricted to the gradient span we refine the simpler construction and apply well-established techniques of random embedding in high-dimensional space. We discuss these modifications briefly in the end of this section and provide the full details in Section 4 and the full version of the paper [1].

### 3.1 A simpler construction

Let us fix $n, d \geq 1$ and parameters $z = (\alpha, \varepsilon, \gamma) \in \{0,1\}^d \times \mathbb{R} \times \mathbb{R}^2 = \mathcal{Z}$, such that $\alpha \in \{0,1\}^d$, $\varepsilon > 0$ and $\gamma_1, \gamma_2 > 0$. Define the hard instance $f_{(6)} : \mathbb{R}^{d+2} \times \mathcal{Z} \to \mathbb{R}$ as follows:

$$f_{(6)}(w; (\alpha, \varepsilon, \gamma)) = g_\gamma(w; \alpha) + \gamma_1 v_\alpha \cdot w + \varepsilon w \cdot e_{d+2} + r(w), \tag{6}$$

where $g_\gamma$, $v_\alpha$ and $r$ are

- $g_\gamma(w; \alpha) := \sqrt{\sum_{i \in [d]} \alpha(i) h_\gamma^2(w(i))}$ with $h_\gamma(a) := \begin{cases} 0 & a \geq -\gamma_2; \\ a + \gamma_2 & a < -\gamma_2, \end{cases}$

- $r(w) := \max\{0, \max_{i \in [d+1]}\{w(i)\}\}$,

- $v_\alpha(i) := \begin{cases} -\frac{1}{2n} & \text{if } \alpha(i) = 0; \\ +1 & \text{if } \alpha(i) = 1; \\ 0 & \text{if } i \in \{d+1, d+2\}, \end{cases}$

and $e_{d+2}$ is the $(d+2)$'th standard basis vector. The distribution we will consider is uniform over $\alpha$. That is, we draw $\alpha \in \{0,1\}^d$ uniformly at random and pick the function $f_{(6)}(w; (\alpha, \varepsilon, \gamma))$.

The parameters $\gamma_1$ and $\gamma_2$ of the construction should be thought of as arbitrarily small. In particular, the term $\gamma_1 v_\alpha \cdot w$ in Eq. (6) should be thought of as negligible, and the first term, $g_\gamma$, is roughly

$$g_\gamma(w; \alpha) \approx \sqrt{\sum_{i \in d} \alpha(i)(\max\{-w(i), 0\})^2}.$$

Another useful property of the construction is the population risk $F(w) = \mathbb{E}_{z \sim D} f_{(6)}(w; z)$ is minimized at $w^\star \approx -e_{d+2}$, with expected loss $F(w^\star) \approx -\varepsilon$. However, as we will see, the choice of the perturbation vector $v_\alpha$ and the term $r(w)$ hinder the learner from observing this coordinate and; the first $\Omega(\varepsilon^{-4})$ queries are constrained to a linear subspace where all the points have a high generalization error due to the expectation of the first term $g_\gamma$.

### 3.2 Analysis

We next state the main lemmas we use, with proofs deferred to the full version of the paper [1]. Given a sample $S$, let us denote $\bar{v} = \frac{1}{n} \sum_{\alpha \in S} v_\alpha$, and

$$\text{span}^1\{u_1, u_2, \ldots\} := \text{span}\{u_1, u_2, \ldots\} \cap \mathcal{W}.$$

Additionally, given a fixed sample we write

$$\mathcal{I}(S) = \{i : \alpha(i) = 0 \ \forall \alpha \in S\} \cup \{d + 1\}$$

for the set of coordinates $i \in [d]$ such that $\alpha(i) = 0$ for every $\alpha$ in the sample $S$, plus the coordinate $d + 1$.

**Lemma 3.1.** *Let $\gamma_1 \leq \frac{1}{2T}$, $\gamma_2 = \frac{2\gamma_1}{\varepsilon}$, and suppose that the sample $S$ satisfies $|\mathcal{I}(S)| > T$. Then there exists a first-order full-batch oracle such that for any algorithm that adheres to*

$$w_t \in \text{span}^1\{\nabla F_S(w_0), \nabla F_S(w_1), \dots, \nabla F_S(w_{t-1})\}, \tag{7}$$

*with respect to $f(w; (\alpha, \varepsilon, \gamma))$ defined in Eq. (6), we have*

$$w_t \in \underset{i \in \mathcal{I}_t(S)}{\text{span}^1} \left\{\gamma_1 \bar{v} + \varepsilon e_{d+2} + e_i\right\} \text{ for all } t \in [T],$$

*where $\mathcal{I}_t(S)$ is the set of the $t + 1$ largest coordinates in $\mathcal{I}(S)$.*

We next observe that in any span of the form $\{\gamma_1 \bar{v} + \varepsilon e_{d+2} + e_i\}_{i \in \mathcal{I}_T(S)}$ such that $|\mathcal{I}_T(S)| \leq T$, we cannot find a solution with better risk than 0. On the other hand, note that for $\bar{w} = -e_{d+2}$, we have that

$$f_{(6)}(\bar{w}; (\alpha, \varepsilon, \gamma)) = -\varepsilon.$$

In other words, our lower bound stems from the following result:

**Lemma 3.2.** *For sufficiently small $\gamma_1 \leq 2n\varepsilon\gamma_2$, $\gamma_2 \leq \varepsilon/\sqrt{4T}$, and any vector $\|\bar{v}\| \leq \sqrt{d}$, any output*

$$w_S \in \underset{i \in \mathcal{I}_T(S)}{\text{span}^1} \{\gamma_1 \bar{v} + \varepsilon e_{d+2} + e_i\},$$

*satisfies*

$$\frac{1}{2}\sqrt{\sum_{i \in [d]} h_\gamma^2(w_S(i))} + \varepsilon w_S(e_{d+2}) \geq \min\left\{1 - 2\varepsilon^2\sqrt{T}, 0\right\} - \frac{1}{2}\varepsilon. \tag{8}$$

**Lower bound proof sketch for span-restricted algorithms of the form** (5). First, observe that the probability of an arbitrary index $i$ to satisfy $\alpha(i) = 0$ for all $\alpha \in S$ is $(1/2)^n$. Therefore, $|\mathcal{I}(S)| - 1$, the number of indexes that hold this from the possible $d$, is distributed as a binomial with $d$ experiments and success probability $p = 2^{-n}$. Using elementary probability arguments one can show that for sufficiently large $d$ we have $|\mathcal{I}(S)| > T$ with high probability; see Claim B.2 in the appendix. This implies that the conditions of Lemmas 3.1 and 3.2 hold w.h.p. To conclude, we relate the LHS of Eq. (8) to the expected risk

$$F(w) = \underset{\alpha \sim D}{\mathbb{E}}[f_{(6)}(w; (\alpha, \varepsilon, \gamma))] = \underset{\alpha \sim D}{\mathbb{E}}[g_\gamma(w; \alpha)] + \gamma_1 \cdot \underset{\alpha \sim D}{\mathbb{E}}[v_\alpha] \cdot w + \varepsilon w \cdot e_{d+2} + r(w).$$

As $g_\gamma(w; \alpha)$ is convex w.r.t. $\alpha$ (since $\alpha(i) = \alpha^2(i)$) we can apply Jensen's inequality with $\mathbb{E}_{\alpha \sim D}[\alpha(i)] = \frac{1}{2}$ to obtain:

$$\underset{\alpha \sim D}{\mathbb{E}}[g_\gamma(w_S; \alpha)] \geq \frac{1}{2}\sqrt{\sum_{i \in [d]} h_\gamma^2(w_S(i))}.$$

Applying the Cauchy-Schwarz inequality to the second term while also using the facts that $\|v_\alpha\| \leq \sqrt{d}$ and that $w_S$ is in the unit ball, we get:

$$\gamma_1 \underset{\alpha \sim D}{\mathbb{E}}[v_\alpha] \cdot w \geq -\gamma_1 \underset{\alpha \sim D}{\mathbb{E}}[\|v_\alpha\| \cdot \|w\|] \geq -\gamma_1\sqrt{d}.$$

For sufficiently small $\gamma_1$ this term is negligible, and since $r(w) \geq 0$ we get that the expected risk is approximately the LHS term in Eq. (8). Lastly, recalling that $F(-e_{d+2}) = -\varepsilon$ we get that

$$F(w_S) - \min_{w \in \mathcal{W}} F(w) \geq \frac{1}{2}\varepsilon + \min\left\{1 - 2\varepsilon^2\sqrt{T}, 0\right\} \text{ w.h.p.}$$

The same lower bound (up to a constant) also holds in expectation by the the law of total expectation. Our distribution is supported on 5-Lipschitz convex functions, so that re-parametrizing $\frac{1}{10}\varepsilon \to \varepsilon$ as well as $f_{(6)}$ yields the claimed lower bound (4) for the case of span-restricted algorithms. ∎

### 3.3 Handling general full-batch algorithms

The above construction establishes an $\Omega(1/\varepsilon^4)$ oracle complexity lower bound on any algorithm whose iterates lie in the span of the previous gradients. While this covers a large class of algorithms, techniques like preconditioning [13], coordinate methods [27] and randomized smoothing [14] do not satisfy this assumption. In fact, a trivial algorithm that always outputs $-e_{d+2}$ will solve the hard instance (6) in a single iteration.

To address general algorithms, we employ a well-established technique in optimization lower bounds [30, 8, 12] wherein we embed a hard instance $f(w; z)$ for span-constrained algorithms in a random high-dimensional space. More concretely, we draw a random orthogonal matrix $U \in \mathbb{R}^{d' \times d}$ ($U^\top U = I_{d \times d}$) and consider the $d' > d$-dimensional instance $f_U(w; z) = f(U^\top w; z)$ along with its corresponding empirical objective $F_{S,U}(w) = \frac{1}{n} \sum_{i \in [n]} f_U(w; z_i)$. Roughly speaking, we show that for a general algorithm operating with the appropriate subgradient oracle for $F_{S,U}$ the iterate $w_t$ is approximately in the span of $\{\nabla F_{S,U}(w_0), \ldots, \nabla F_{S,U}(w_{t-1})\}$ in the sense that the component of $w_t$ outside that span is nearly orthogonal to the columns of $U$. Consequently, the response of the oracle to the query $w_t$ at iteration $t$ is, with high probability, identical to the information it would return if queried with the projection of $w_t$ to the span of the previously observed gradients. This reduces, in a sense, the problem back to the span-restricted setting described above.

For the embedding technique to work, we must robustify the hard instance construction so that small perturbations around points in the span of previous gradients do not "leak" additional information about the embedding $U$. To do that we make a fairly standard modification to the component $r(w)$ in (6) (known as Nemirovski's function [12, 7]), replacing it with $\max\{0, \max_{i \in [d]}\{w(i) + i\gamma'\}, w(d+1) + \gamma''\}$, where $\gamma', \gamma''$ are small offset coefficients that go to zero as the embedding dimension $d'$ tends to infinity. We provide the full construction and the proof of Theorem 2.1 in Section 4 and the full version of the paper [1].

## 4 The Full Construction

As explained above, the key difference between the simplified construction $f_{(6)}$ and the full construction with which we prove Theorem 2.1 is that we modify the Nemirvoski function term $r(w)$ in order to make it robust to queries that are nearly within a certain linear subspace. In particular, we bias the different terms in the maximization defining $r(w)$ so as to control the index of the coordinate attaining the maximum. For ease of reference, we now provide a self-contained definition of our full construction with the modified Nemirovski function.

Fix $n, d \geq 1$ and parameters $z = (\alpha, \varepsilon, \gamma) \in \{0, 1\}^d \times \mathbb{R} \times \mathbb{R}^3 = \mathcal{Z}$ are such that $\alpha \in \{0, 1\}^d$, $\varepsilon > 0$ and $\gamma_1, \gamma_2, \gamma_3 > 0$. Define the hard instance $f_{(9)} : \mathbb{R}^{d+2} \times \mathcal{Z} \to \mathbb{R}$ as follows:

$$f_{(9)}(w; (\alpha, \varepsilon, \gamma)) = g_\gamma(w; \alpha) + \gamma_1 v_\alpha \cdot w + \varepsilon w \cdot e_{d+2} + r(w), \tag{9}$$

where $g_\gamma, v_\alpha$ and $r$ are

- $g_\gamma(w; \alpha) := \sqrt{\sum_{i \in [d]} \alpha(i) h_\gamma^2(w(i))}$  with  $h_\gamma(a) := \begin{cases} 0 & a \geq -\gamma_2; \\ a + \gamma_2 & a < -\gamma_2, \end{cases}$

- $r(w) := \max\{0, \max_{i \in [d+1]}\{w(i) + \sigma_i\}\}$  with  $\sigma_i := \begin{cases} i \cdot \frac{\gamma_1 \gamma_3}{4dn} & \text{if } i \in [d]; \\ 2\gamma_3 & \text{if } i = d+1. \end{cases}$

- $v_\alpha(i) := \begin{cases} -\frac{1}{2n} & \text{if } \alpha(i) = 0; \\ +1 & \text{if } \alpha(i) = 1; \\ 0 & \text{if } i \in \{d+1, d+2\}, \end{cases}$

and $e_i$ is the $i$'th standard basis vector in $\mathbb{R}^{d+2}$. We consider a distribution $D$ over $\alpha$ that is distributed uniformly over $\{0, 1\}^d$; that is, we draw $\alpha \in \{0, 1\}^d$ uniformly at random and pick the function $f_{(9)}(w; (\alpha, \varepsilon, \gamma))$. The rest of the parameters are set throughout the proof as follows:

$$\gamma_1 = \frac{\varepsilon \gamma_2}{4}, \quad \gamma_2 = \frac{\varepsilon}{T\sqrt{d}}, \quad \gamma_3 = \frac{\varepsilon}{16}. \tag{10}$$

With this choice of distribution $D$ as well our choice of parameters we obtain, since $\|v_\alpha\| \le \sqrt{d}$ and by our choice of $\gamma_1$ (as well as Jensen's inequality and $r(\cdot) \ge 0$):

$$F(w) = \underset{\alpha \sim D}{\mathbb{E}} \left[ f_{(9)}(w; (\alpha, \varepsilon, \gamma)) \right] \ge \frac{1}{2} \sqrt{\sum_{i \in [d]} h_\gamma^2(w(i))} + \varepsilon w(d+2) - \frac{\varepsilon}{4}. \tag{11}$$

Notice that we also have that for a choice $w^\star = -e_{d+2}$, since $r(w^\star) = 2\gamma_3$:

$$F(w^\star) = -\varepsilon + \frac{\varepsilon}{8} = -\frac{7\varepsilon}{8} \tag{12}$$

Our development makes frequent use of the following notation from Section 3:

$$\mathcal{I}(S) = \{i : \alpha(i) = 0 \ \forall \alpha \in S\} \cup \{d+1\}, \ \mathcal{I}_t(S) = t \text{ largest elements in } \mathcal{I}(S), \text{ and } \bar{v} = \frac{1}{n} \sum_{\alpha \in S} v_\alpha.$$

We begin with the following lemma, which is a robust version of Lemma 3.1 in Section 3. The proof is provided in the full version of the paper [1].

**Lemma 4.1.** *Suppose that $w_0 = 0$. Consider $f_{(9)}(w; (\alpha, \varepsilon, \gamma))$ with parameters as in Eq. (10). Suppose $S$ is a sample such that $|\mathcal{I}(S)| > t + 1$. Assume that $w$ is such that*

$$w = w_t + q,$$

*where*

$$w_t \in \underset{i \in \mathcal{I}_t(S)}{\text{span}^1} \left\{ \gamma_1 \bar{v} + \varepsilon e_{d+2} + e_i \right\}, \text{ and } \|q\|_\infty \le \min\left\{ \frac{\gamma_2}{3}, \frac{\gamma_1 \gamma_3}{16dn} \right\}. \tag{13}$$

*Then,*

$$\nabla F_S(w) = \gamma_1 \bar{v} + \varepsilon e_{d+2} + e_i,$$

*for some $i \in \mathcal{I}_{t+1}(S)$, where $\mathcal{I}_t(S)$ is the set of the $t+1$ largest coordinates in $\mathcal{I}(S)$.*

The following corollary states that the gradient oracle's answers are resilient to small perturbation of the query (as long as they are in vicinity of the "right" subspace): the proof is provided in the full version of the paper [1]:

**Corollary 4.2.** *Assume that $w$ is such that*

$$w = w_t + q,$$

*where*

$$w_t \in \underset{i \in \mathcal{I}_t(S)}{\text{span}^1} \left\{ \gamma_1 \bar{v} + \varepsilon e_{d+2} + e_i \right\}, \text{ and } \|q\|_\infty \le \frac{1}{4\sqrt{d}} \min\left\{ \frac{\gamma_2}{3}, \frac{\gamma_1 \gamma_3}{16dn} \right\}. \tag{14}$$

*Then,*

$$\nabla F_S(w) = \nabla F_S(\Pi_{t+1}(w)), \qquad F_S(w) = F_S(\Pi_{t+1}(w)),$$

*where $\Pi_t$ is a projection onto $\text{span}_{i \in \mathcal{I}_t(S)} \{ \gamma_1 \bar{v} + \varepsilon e_{d+2} + e_i \}$.*

## Acknowledgements and Disclosure of Funding

This work has received support from the Israeli Science Foundation (ISF) grant no. 2549/19 and grant no. 2188/20, from the Len Blavatnik and the Blavatnik Family foundation, from the Yandex Initiative in Machine Learning, and from an unrestricted gift from Google. Any opinions, findings, and conclusions or recommendations expressed in this work are those of the author(s) and do not necessarily reflect the views of Google.

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
