# OpenReview forum: "Never Go Full Batch (in Stochastic Convex Optimization)"
_NeurIPS.cc/2021/Conference — NeurIPS 2021 Poster_

### Official Review · Reviewer_ge1M · 2021-07-09

**Rating:** 7
**Confidence:** 2

**Summary:**

In this paper the authors prove an error lower-bound for full-batch first-order algorithms. This is a generalization of the bound given in Bassily et al. [2]. This new bound of the authors
- is true for a wider class of functions : GD and SGD, non-smooth and convex
- implies that to get an $\epsilon$-risk, full-batch methods require at least $\Omega (1/\epsilon^4)$ iterations
This shows the advantage of stochastic methods like SGD which only require $\Omega (1/\epsilon^2)$ iterations to reach the same risk.

**Limitations And Societal Impact:**

This paper is theoretical only. No experiments are provided, even to illustrate the different bounds.

This paper is a bit out of my standard scientific scope.

**Main Review:**

The authors clearly explain the background of Stochastic Convex Optimization (SCO) and present existing results and bounds. The authors also underline the fact that their bound is valid for a large family of algorithms (GD, projected GD, with smooth or non-smooth regularization, etc).
To extend their results to general full-batch algorithms, they apply a classical technique described in section 4.3 where they use a random high-dimensional embedding and apply the previous bound on a modified objective.

**Comment after rebuttal**

In view of the reviews and the discussions, the authors clarified their technical contribution compared to [1] and presented how this generalization could be applied to many 1st order methods including Nesterov's acceleration, CG etc.

Thus, I will increase my score to 7.

**Time Spent Reviewing:**

7

---

> ### Author Response · Authors · 2021-08-10
> **Thanks for the review -- Author's response**
>
> Thank you for the review and the overall positive assessment.
>
> > “This paper is theoretical only. No experiments are provided, even to illustrate the different bounds”
>
> Our paper is indeed purely theoretical and its main result is a lower bound that applies to any algorithm---as we do not make any claims beyond this lower bound, we feel that experiments will not make our paper stronger.  Moreover, as we discuss in Section 1.1, our findings on the limitations of full-batch methods seem to align with existing empirical observations.

---

> > ### Comment · Reviewer_ge1M · 2021-08-19
> > **"Wider family of algorithms"**
> >
> > Dear authors,
> >
> > Thank you for your clear answer.
> >
> > I have an additional question concerning the extensions of your work. Line 186 it is mentioned that your main result, that is Thm 3.1, extends Thm 3.1 from [1] which only deals with GD updates, "to an even wider family of algorithms". What kind of other algorithms do satisfy (3) except GD ?
> >
> >
> > Minor info: typo line 311 "oracle(independent" : missing space

---

> > > ### Author Response · Authors · 2021-08-19
> > > **Wider family of algorithms**
> > >
> > > Thank you for the question, we are happy to clarify.  The following examples include algorithms that were not addressed in previous results:
> > > - GD with any (possibly adaptive) sequence of learning rates;
> > > - GD on a regularized empirical risk (with any regularization penalty and any learning rate);
> > > - noisy GD and GD with dropout.
> > > This is discussed in lines 57-63, but see also the “Examples” paragraph in Section 2.
> > >
> > > Notice that the focus of this paper is generalization, so above we listed several examples of algorithms and techniques that are often relied upon to improve generalization (yet our result implies they are ultimately inferior to stochastic methods like SGD).  Nevertheless, our result generalizes to **any first-order method applied on the empirical risk**: conjugate gradient, quasi-newton methods (e.g. L-BFGS), trust-region methods (e.g. Levenberg-Marquardt), conditional gradient methods (e.g. Frank-Wolfe), and so on.

---

> > > > ### Comment · Reviewer_ge1M · 2021-08-19
> > > > **Thank you for the precision**
> > > >
> > > > Thank you for detailing 1st order methods encompassed by your assumptions.

---

### Official Review · Reviewer_s3Yy · 2021-07-09

**Rating:** 6
**Confidence:** 3

**Summary:**

This paper considers a problem where one is trying to find an input that minimizes the expected value of a function that one has sample values of. Its goal is to show that algorithms that only have the ability to query the gradient of the average of the samples at specified points may need more queries than a function that can check the gradient on individual samples. In order to do that, the paper finds a class of probability distributions over functions for which any algorithm that gains information on the functions only by querying the gradient of the average of the sampled function needs $\Omega(1/\epsilon^4)$ queries to get within $\epsilon$ of the minimum expected loss.

**Limitations And Societal Impact:**

Yes

**Main Review:**

As far as I can tell, this never actually proves that one can get a loss within $\epsilon$ of the optimum in $o(1/\epsilon^4)$ queries if one is not restricted to working with the averaged gradient, which means it does not actually prove the separation result in the abstract. The lower bound on the number of queries needed to get near the minimum is much less interesting without the contrast than it would be with it.

This appears to mostly use known techniques to prove the bound. The idea of bounding what one can learn from the gradient reminds me of statistical query bounds, although those would be weaker in that they would use a distorted form of the gradient. The organization is mostly fine, although separating the lemmas in section 4 from their proofs is confusing.

----------------------------------edit--------------------------

The textbook the authors mention in their rebuttal contains the positive half of the separation. So, I am now recommending that this paper be accepted.

**Time Spent Reviewing:**

9

---

> ### Author Response · Authors · 2021-08-10
> **Thanks for the review -- Author's response**
>
> Thank you for the review---below we address the specific concerns raised:
>
> > “This never actually proves that one can get a loss within $\epsilon$ of the optimum in $o(1/\epsilon^4)$ queries if one is not restricted to working with the averaged gradient, which means it does not actually prove the separation result in the abstract”
>
> **We do prove the claimed separation between full-batch methods and algorithms such as stochastic gradient descent as stated in the abstract**.  It is a classical result in ML/optimization that $O(1/\epsilon^2)$ iterations of SGD reach $\epsilon$ generalization loss. (This result and its proof can be found in any standard textbook, e.g. “Understanding Machine Learning" by Shalev Shwartz and Ben David). Hence, there exists a method (SGD) which is not restricted to using the averaged gradient and outperforms full-batch methods. This fact is mentioned briefly in the abstract, and noted in passing throughout the introduction, as well as in the discussion after the main result.  We do agree that this should be better highlighted and cited, and we will do so in the revision.
>
> > “This appears to mostly use known techniques to prove the bound”
>
> Our proof naturally builds upon previous ideas, and we were as transparent as possible about this in our exposition and gave credit where credit is due. That said, we strongly disagree that most of our proof is an application of known techniques.  The standard approach to establish generalization gaps is via a reduction to either the statistical rate or optimization rate; both methods can at best yield bounds of order $\Omega(1/\epsilon^2)$ which is both the statistical and the optimization minimax rate.  So these existing techniques are inapplicable for our purpose.  The only exception is [1], a work which indeed we build upon (and mention this very clearly), but while they provide an analysis for a specific instance of GD by a concrete analysis of its trajectory, we provide a general argument that captures a much wider class of algorithms than does [1].
>
> > “separating the lemmas in section 4 from their proofs is confusing”
>
> Thanks for pointing out---we will improve the flow in the revision.

---

### Official Review · Reviewer_aT3D · 2021-07-15

**Rating:** 6
**Confidence:** 4

**Summary:**

This work studies the generalization performance of algorithms in stochastic convex optimization. Specifically, the authors study the full-batch algorithms where only full gradient and function value information are accessible. By constructing a hard instance of the objective function, the authors show that any full-batch algorithm needs at least $\Omega(1/\epsilon^4)$ iterations to obtain an $\epsilon$-risk function, while stochastic algorithms such as SGD only need $O(1/\epsilon^2)$.

**Limitations And Societal Impact:**

The authors discuss the limitations and there is no negative societal impact about their work.

**Main Review:**

This paper shows that there exists a separation between full-batch algorithms and stochastic algorithms in the generalization performance of empirical minimization. Such a separation has been well studied in the pure optimization setting, and this work is the first to study such a problem in the generalization setting. The presentation is clear and easy to understand. The proof is technically sound. Some of my comments are as follows.

- It is a bit hard to tell the significance of the proposed result. On the one hand, the lower bound result reveals that full-batch algorithms will be sample-inefficient compared with SGD, which is not studied in previous works. On the other hand, such a phenomenon is not surprising, since similar separation results have already been studied in previous works studying optimization errors. The tool the authors used is standard (hard instance by  Nemirovski et. al), and it is not clear how the authors contribute given existing works.

- For the use of random rotation matrices, there are some additional works studying optimization lower bounds which also use them. The authors may want to highlight these works in the main text.

[1] Fang, C., Li, C. J., Lin, Z., & Zhang, T. (2018). Spider: Near-optimal non-convex optimization via stochastic path integrated differential estimator. arXiv preprint arXiv:1807.01695.

[2] Zhou, D., & Gu, Q. (2019, May). Lower bounds for smooth nonconvex finite-sum optimization. In International Conference on Machine Learning (pp. 7574-7583). PMLR.

[3] Arjevani, Y., Carmon, Y., Duchi, J. C., Foster, D. J., Srebro, N., & Woodworth, B. (2019). Lower bounds for non-convex stochastic optimization. arXiv preprint arXiv:1912.02365.

**Time Spent Reviewing:**

5 hrs

---

> ### Author Response · Authors · 2021-08-10
> **Thanks for the review -- Author's response**
>
> Thank you for the review and the positive assessment---below we address the specific concerns raised:
>
> > “the lower bound result reveals that full-batch algorithms will be sample-inefficient compared with SGD”
>
> We do not show sample-inefficiency but rather iteration inefficiency; in fact, as we discuss in the paper, the sample complexity of GD and SGD is comparable.
>
> > “similar separation results have already been studied in previous works studying optimization errors”
>
>
> We respectfully disagree. When studying optimization (training) errors, no separation between full-batch and stochastic gradient descent exists: both methods have iteration complexity $\Theta(1/\epsilon^2)$.  In other words, full-batch GD and SGD decrease the *training error* at the same rate (in terms of number of iterations), but as we prove, any full-batch method will decrease the *test error* much more slowly (in the worst-case) than SGD.
>
> > “The tool the authors used is standard (hard instance by Nemirovski et. al), and it is not clear how the authors contribute given existing works”
>
>
> Again, our separation result does not involve lower bounding the training/optimization error.  Nemirovski's function is indeed an ingredient in our construction, but it fulfills a totally different role in our lower bound than it does in classical (training/optimization) lower bounds.  In our construction, the challenge faced by the learner is not to optimize Nemirovski's function (which can be done with $O(1/\epsilon^2)$ iterations). Instead, the Nemirovski term is used as a "red herring"---it simply generates irrelevant gradients, and the learner's task is to identify the direction of a small linear term that is being drowned out by these irrelevant gradients.
>
>
> > “The use of random rotation matrices”
>
> Thank you for pointing out these references, we will cite accordingly.

---

### Official Review · Reviewer_XAZp · 2021-07-21

**Rating:** 6
**Confidence:** 4

**Summary:**


The paper provides a lower bound on the iteration complexity of any full batch algorithm in stochastic convex optimization. In particular, they show that for every T, there exists a function (which depends on T) for which the suboptimality in terms of excess risk for any full batch algorithm run for T steps is lower bounded by 1/T^{1/4}. The proof technique extends upon the lower bound construction of Amir et. al. 2021 [1] for the iteration complexity of GD algorithm, and is based on a span based argument.


**Main Review:**

I support accepting the paper. The main reasons for a low score are:

(1. ) SGD vs ERM: We already know from earlier works that Empirical risk minimizing (ERM) solutions may fail to have low suboptimality at the population level (for eg. in [14], [25]).  Full batch descent algorithms find the ERM solution. I do not see a lot of motivation in proving lower bounds for algorithms which go towards solutions which we know may not generalize.

(2.) Iteration complexity vs work: We know that SGD is statistically optimal for stochastic convex optimization. Furthermore, it is also optimal work wise. It is easy to see that running event 1 iteration of GD is as expensive as running SGD to convergence. Why should we care about lower bounds for full batch algorithms for which we already know that even 1 iteration is worse than the optimal and easy to implement algorithm.

(3.) GD with projection vs without projection: In the paper, the authors run GD with projection and compare the suboptimality of the returned solution with the minimizer in the unit norm ball. In practice, we generally run SGD / GD without projection? Is there any hope to extend the lower bounds for full batch methods run without the projection step.

(4.) Statement of the results:

(a) The authors should add in the main body that the function depends on T (as the parameters and the problem dimension are chosen depending on the value of T).

(b) I am confused by the way the result is presented in Theorem 3.1. If we look at the proof at the end of page 9, we see that there is an extra \beta-dependent term multiplying (1/4 - \eps^2 T) which is data dependent. This term can be arbitrarily small and has been hidden in the proof of Theorem 3.1. One can simply get rid of this term by setting \eps = 1/T^{1/4} which the authors should do. This would just give a lower bound of 1/T^{1/4} in Theorem 3.1.

The paper is well written otherwise. It might be useful to explicitly state the upper / lower bounds on \gamma_1 and \gamma_2 in various lemmas in the main body.


**Time Spent Reviewing:**

8-10 hours

---

> ### Author Response · Authors · 2021-08-10
> **Thanks for supporting acceptance**
>
> Thank you for the thorough review and the positive assessment!  We hope that the discussion below (that we will integrate into our final version) will help you in raising your score so that it reflects your recommendation of acceptance, as borderline scores are often perceived as recommendation for rejection.
>
>
> > “Full batch descent algorithms find the ERM solution. I do not see a lot of motivation in proving lower bounds for algorithms which go towards solutions which we know may not generalize”
>
>
> It is not very accurate to say that "full batch descent algorithms find the ERM solution". For example, as we discuss in the paper, GD over the regularized empirical risk is a full-batch method yet it does not converge towards the ERM solution (but rather to the regularized ERM solution which in fact is known to generalize [25]).
>
> Moreover, as we also discuss, full-batch GD that takes $O(1/\epsilon^4)$ steps is actually capable of generalizing, even without regularization, despite “going towards” the ERM solution. So it seems that **generalization is not so much a question of towards where an algorithm converges to, and it is crucial to appraise the rate in which it converges and the trajectory it assumes**.
>
> > “Why should we care about lower bounds for full batch algorithms for which we already know that even 1 iteration is worse than the optimal and easy to implement algorithm”
>
> It is true that in terms of total “work” full-batch methods are known to be inferior to SGD, though even in these terms we show a lower bound of $\Omega(n^3)$ work for full-batch GD, which improves on the naive bound $\Omega(n^2)$.  However, we disagree that one should only care about the total work and not about the iteration complexity:
>
> First, full-batch GD, as well as its regularized variants, are very basic algorithms and understanding their generalization properties is a fundamental task.  In particular, one takeaway from our result is that the implicit regularization effect of the sub-sampling used in SGD (which is absent from full-batch methods) is crucial for generalizing efficiently, even when large batches are completely free to compute.
>
> Second, the “work” of an algorithm is not necessarily well-defined and depends strongly on e.g. the computing platform: for example, in a distributed architecture (very common in large scale ML) “work” is split between worker machines, and the runtime per step more accurately represents the time to compute the full-batch gradient.  Our results demonstrate that even in this case, where full-batch gradients “come for free”, using them can harm the solution quality quite significantly.
>
> > GD with projection vs without projection:
>
> Our result can be extended to allow querying from an arbitrary large bounded domain.  However, since it requires technically-involved (and not very interesting) modifications to the analysis, we preferred to focus on a simpler setting for the sake of clarity.  SCO over a bounded convex domain is a classical and well-studied setting that warrants this exclusive focus.
>
> > “The authors should add in the main body that the function depends on T”
>
> Thanks for pointing out---we will make sure to do so in the revision.
>
> > “there is an extra $\beta$-dependent term multiplying $(1/4 - \epsilon^2 T)$ which is data dependent. This term can be arbitrarily small and has been hidden in the proof of Theorem 3.1”
>
> Note that the term inside the $\Omega(\cdot)$ in Theorem 3.1 is non-positive (since it is the minimum between something and zero). Therefore, in page 9 we only need an upper bound on the coefficient of $\frac{1}{4} - \epsilon^2 \sqrt{T}$, which we establish immediately below.
> If there are remaining concerns we will be happy to further clarify during the discussion.
>
> > “It might be useful to explicitly state the upper / lower bounds on \gamma_1 and \gamma_2 in various lemmas”
>
> Indeed---will be taken care of.  Thanks.

---

### Decision · Program_Chairs · 2021-09-27

**Decision:**

Accept (Poster)

**Comment:**

During the discussion phase, all reviewers and the AC found the results are interesting and novel enough for a presentation at NeurIPS. The paper has presented some solid results and bring new insights about negative results of full batch methods.